# Learning to Transfer Heterogeneous Translucent Materials from a 2D Image to 3D Models

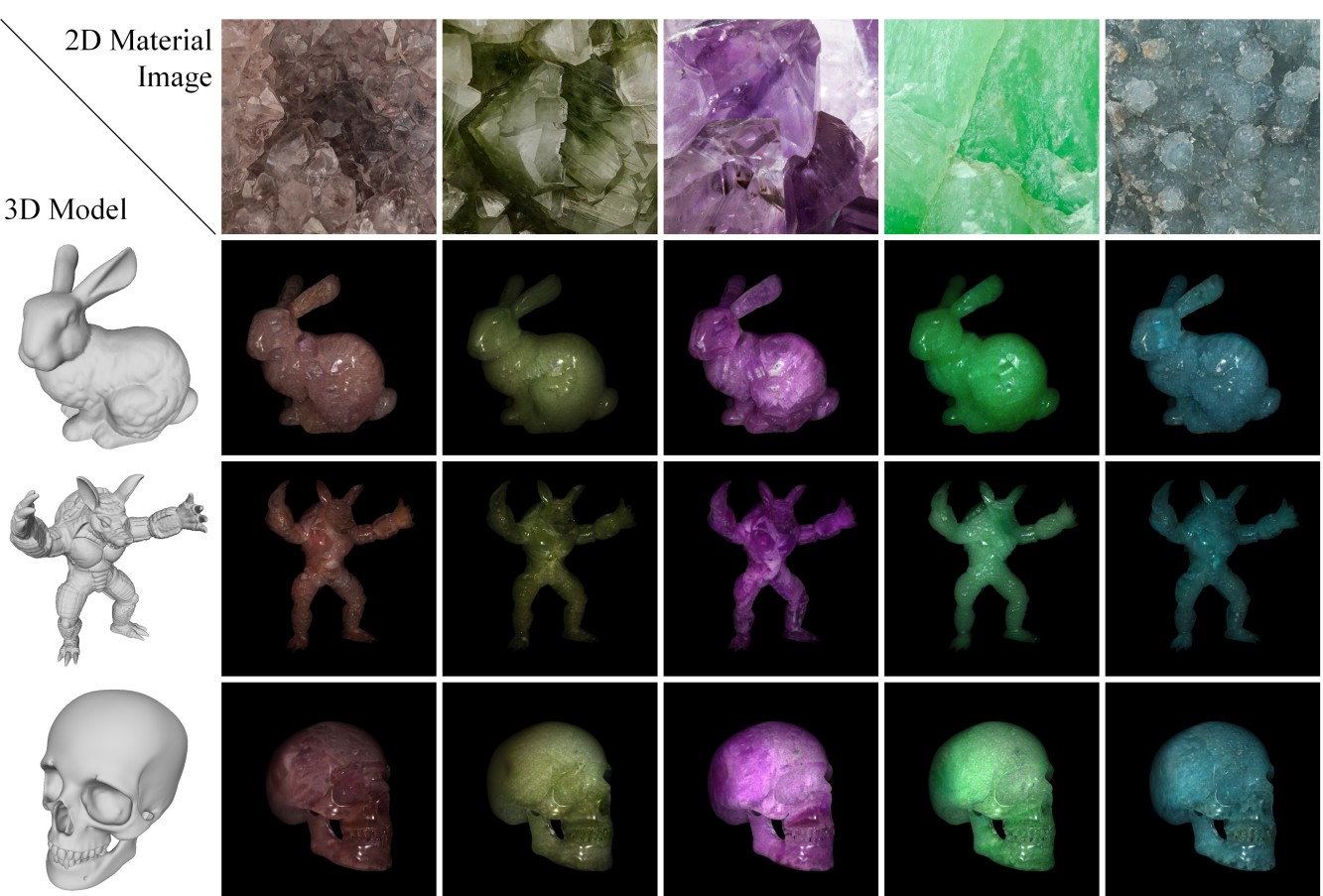

**Figure 1: Tranfer heterogeneous translucent materials from single images to 3D models. Our method achieves realistic material transfer on different models based on different 2D images.**

## ABSTRACT

Great progress has been made in rendering translucent materials in recent years, but automatically estimating parameters for heterogeneous materials such as jade and human skin remains a challenging task, often requiring specialized and expensive physical measurement devices. In this paper, we present a novel approach for estimating and transferring the parameters of heterogeneous translucent materials from a single 2D image to 3D models. Our method consists of four key steps: (1) An efficient viewpoint selection algorithm to minimize redundancy and ensure comprehensive coverage of the model. (2) Initializing a homogeneous translucent material to render initial images for translucent dataset. (3) Edit the rendered translucent images to update the translucent dataset. (4) Optimize the edited translucent results onto material parameters using inverse rendering techniques. Our approach offers a practical and accessible solution that overcomes the limitations of existing methods, which often rely on complex and costly specialized devices. We demonstrate the effectiveness and superiority of our proposed method through extensive experiments, showcasing its ability to transfer and edit high-quality heterogeneous translucent materials on 3D models, surpassing the results achieved by previous techniques in 3D scene editing.

## CCS CONCEPTS

• **Computing methodologies → Appearance and texture representations**; • **General and reference → General conference proceedings**.

## KEYWORDS

translucent materials, heterogeneous, material editing, differentiable rendering, style transfer

# 1 INTRODUCTION

Translucent materials, exhibiting high scattering characteristics, are ubiquitous in our daily lives, ranging from precious jewels to the intricate cellular structures found in biological organisms. These materials play vital roles across a multitude of applications. The rendering of such translucent materials has witnessed remarkable progress through physically based rendering techniques [33] , including path tracing [20] and volumetric path tracing [32] models, seamlessly integrated into rendering engines like Mitsuba3 [18], among others.

While rendering materials have matured, parameter estimation for these materials has historically involved manual adjustments [18, 31], consuming substantial time and financial resources. Efforts to automate this process have been ongoing [6, 8, 13, 24, 25], but they are limited to homogeneous materials and struggle to extend to heterogeneous ones like jade, marble, or human skin. The latter is characterized by heterogeneity, surface microstructure, and short scattering mean free paths, making manual estimation impractical.

Addressing the challenge of parameter estimation for heterogeneous translucent materials, some researchers resort to physical measurement instruments [9, 14, 37, 41, 43, 44]. However, this approach is specialized and costly, limiting its accessibility for ordinary consumers or general computer graphics applications.

Recently, InverseTranslucent [5] introduced a 3D reconstruction method for translucent objects using low-cost handheld acquisition setups. This approach effectively estimates material parameters through integrating multi-view images and a translucent differentiable renderer. Nevertheless, limitations exist, including the need for fixed lighting conditions during image acquisition and the manual collection of multi-view images, restricting their applicability.

In this paper, we propose a novel method for transferring heterogeneous translucent materials from a single image. As illustrated in Figure 1, our approach enables the direct estimation of relevant parameters under natural lighting conditions and seamlessly transfers the material onto a 3D model. To begin, we strategically select sparse viewpoints that minimize redundancy while ensuring comprehensive coverage of the model. Utilizing a translucent renderer, we render translucent initial images from the selected sparse viewpoints. Subsequently, we perform material transfer sequentially on single-view rendered images, effectively transferring the heterogeneous translucent material to the initial image. This approach successfully preserving the structure and light properties of the initial image while accurately transferring the heterogeneous translucent materials. Inspired by the work of Instruct-NeRF2NeRF [15], we introduce an innovative iterative editing-optimization strategy. By iteratively editing and performing inverse rendering, we update the edited translucent results onto the 3D model, ensuring consistency and coherence in our multi-view material editing results. In sum, our contributions include:

- Proposal of a method for transferring heterogeneous translucent materials based on a single image.
- Design of a translucent material transfer editor preserving both lighting and structure, enabling high-quality material transfer for 3D models with initialized translucent materials.

- Introduction of an iterative editing-optimization strategy ensuring consistency in material editing for heterogeneous translucent materials across multiple viewpoints.

# 2 RELATED WORK

## 2.1 Translucent Rendering

Achieving realistic translucent rendering involves simulating the scattering and transmission of light through the material. The Bidirectional scattering surface reflectance distribution function (BSS-RDF) is generally used to simulate subsurface scattering effects. Jensen et al. [19] proposed a practical dipole model that can be effectively used in sampling techniques for conventional ray tracers to represent materials scattered by homogeneous subsurfaces. Donner et al. [8] combines photon tracing with diffusion to efficiently render highly scattering translucent materials, and also accounting for internal blockers, complex geometry, translucent inter-scattering, and transmission and refraction of light at the boundary causing internal caustics. d'Eon et al. [6] presents a new analytic BSSRDF for scattering within multilayer translucent materials with arbitrary levels of absorption and under all-frequency illumination which creates accurate results under high-frequency illumination. Vicini et al. [42] proposed a new shape-adaptive BSS-RDF model that retains the efficiency of prior analytic methods while greatly improving overall accuracy. InverseTranslucent [5] accounts for both surface reflection and subsurface scattering to represent translucency using a BSSRDF model. Neural rendering is gaining increasing popularity due to its exceptional fitting capabilities. Suhail et al. [39] combines the strengths of classical light field rendering and geometric reconstruction methods, learning to accurately represent view-dependent effects like translucency from a sparse set of views by operating on a 4D light field representation. Yu et al. [45] proposed Object-Centric Neural Scattering Functions (OSFs) for learning to reconstruct the appearance of opaque and translucent objects.

## 2.2 Differentiable Rendering

Simulating the appearance of translucent materials requires accurate physical parameters. However, obtaining physically accurate parameters for scattering materials remains a challenging task. Differentiable rendering algorithms strive to estimate partial derivatives of pixels in a rendered image with respect to scene parameters, which is difficult because visibility changes are inherently non-differentiable. Certain methods [16, 29, 47] employ path tracing based on Monte Carlo estimation, and by enabling differentiable rendering, they obtain physically-based material estimates from real images. Neural-PBIR [40] introduce a neural material and lighting distillation stage to achieve high-quality predictions for material and illumination and perform physics-based inverse rendering (PBIR) to refine the initial results and obtain the final high-quality reconstruction. For translucent materials, Gkioulekas et al. [13] combine stochastic gradient descent with Monte Carlo rendering and a material dictionary to invert the radiative transfer equation and measure scattering properties. Then, Gkioulekas et al. [12] tackling the problem of heterogeneous inverse scattering from simulated measurements of different computational imaging configurations. InverseTranslucent [5] uses a differentiable subsurface

scattering renderer to represent translucency with a heterogeneous BSSRDF, leveraging low-cost handheld acquisition setups. Li et al. [25] used a physically-based renderer and a neural renderer to estimates homogeneous subsurface scattering parameters from only a pair of captured images of a translucent object.

## 2.3 Material Editing

Automated manipulation of surface materials in 3D models plays a vital role in 3D modeling, enabling users to modify and control the appearance of the models' surfaces. Khan et al. [22] present a method for automatically replacing one material with another, completely different material with only a single high dynamic range image as input.The transformations range from applying a texture to the surface of an object, to the application of any arbitrary BRDF. Subedit [38] decouples the BSSRDF non-local scattering effect into the product of two local scattering profiles, enabling a method for editing heterogeneous subsurface scattering materials obtained from real-world samples. Liu et al. [26] proposes an end-to-end network for image-based material editing, replicating the forward image formation process. Diffusion models are becoming increasingly popular in 3D generation. By combining diffusion models and differentiable rendering, text-guided material generation and editing have become highly effective [27, 35]. Unlike the explicit representation of the surface in 3D models, NeRF can implicitly represent models or scenes. Instruct-Nerf2Nerf [15] propose a method for editing NeRF scenes using text instructions.

Transferring materials using images is a challenging task. For image-based material editing, editing a single image becomes a crucial step. Given a reference image, convolutional neural networks (CNNs) can be used to transfer the style from the reference image to a content image [11, 30]. Inspired by style transfer, StyleGAN [21] generates images by manipulating latent vectors. Hyperstyle [1] inverts real images into the latent space, enabling real image editing. Recently, large-scale text-driven generative models have received widespread attention for their ability to generate highly diverse images based on given text prompts. Prompt-to-Prompt [17] controls editing solely through text, enabling local edits by replacing words. VCT [4] utilizes inversion techniques with a reference image to translate visual concepts while preserving the source image's content. InstructPix2Pix [2] is a method for instructional image editing that can apply specific styles to the edited image.

## 3 METHODS

The methodological overview is presented in Figure 2. First, the Efficient Viewpoints Selection (Section 3.2) method is employed to select multiple viewpoints that capture images of the 3D model, ensuring the selected viewpoints minimize redundancy and cover the entire surface of the model. Subsequently, a translucent material is initialized homogeneously for the 3D model. We use translucent differentiable renderer (Section 3.3) to render initial images from the selected viewpoints, and these images constitute the initial translucent dataset. After initialization, an iterative process is then undertaken, involving an image editing process and an inverse rendering process, to update the translucent dataset and the translucent materials:

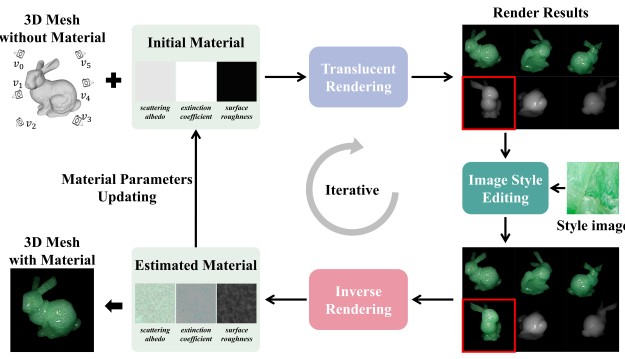

**Figure 2: An Overview of our method. Our method achieves translucent material editing through iterative updates of the translucent dataset: (1) Efficient viewpoint selection for rendering. (2) Translucent material and dataset initialization. (3) Editing translucent rendered images and updating the translucent dataset. (4) Inverse rendering of translucent dataset images to optimize material parameters.**

The image editing process (Section 3.4) utilizes the translucent initial images as the structure and lighting conditions, and a style image as the style and texture conditions. A proposed structure-preserving translucent editor is employed to edit the current material renderings, and the edited translucent results are used to update the translucent dataset. The preliminaries (Section 3.1) provide the techniques used in the editing model.

The inverse rendering process randomly selects images from the translucent dataset as the supervision, and then optimizes the material parameters by performing inverse rendering through the translucent differentiable renderer.

Through the iterative process of our consistent translucent material update (Sec.3.5), we alternately editing and inverse rendering to update our translucent materials. The iterative process preserves the structure and lighting by using the initial image and the current image as conditions, allowing for the consistent transfer of the style image to the translucent material.

## 3.1 Preliminaries

**DDIM sampling:** Text-guided diffusion models have beeb a widely researched area. The primary objective of text-guided diffusion models is to, given a random noise vector $z_T$, denoise $z_T$ conditioned on a given text $P$, until obtaining an output image latent $z_0$ that is close to the description in $P$. To this end, the network $\epsilon_\theta$ is trained to predict the added noise, following the optimization objective:

$$\min_\theta E_{z_0,\varepsilon,t} \, \|\varepsilon - \varepsilon_\theta(z_t, t, C)\|_2^2, \varepsilon{\sim}N(0,I), t{\sim}\text{Uniform}(1,T), \quad (1)$$

where $C$ represents the conditional embedding of $P$, and $z_t$ is the noisy sample with noise added according to the timestamp $t$ on the original sample $z_0$. During inference, for a given noise vector $z_T$, the pre-trained network is used to sequentially predict the noise $\epsilon_\theta$ and remove it through $T$ steps, ultimately generating a clear image.

The Diffusion Denoise Implicit Model (DDIM) [36] sampling is employed to generate images, with the computation formula:

$$z_{t-1} = \sqrt{\frac{\alpha_{t-1}}{\alpha_t}} z_t + \left( \sqrt{\frac{1}{\alpha_{t-1}} - 1} - \sqrt{\frac{1}{\alpha_t} - 1} \right) \cdot \varepsilon_\theta(z_t, t, C), \quad (2)$$

where $\alpha_t$ is the noise scaling factor sequence defined by the diffusion process, which describes the relative strength of the noise at each step as it transitions from a completely noisy state to the original data distribution.

**Classifier-free Guidance:** Classifier-free Guidance (CFG) is a key technique to control the influence of the text condition on the image generation process, without relying on a separate text classifier model. CFG introduces an additional unconditional prediction $\varepsilon_\theta(z_t, t, \varnothing)$, which $\varnothing$ represents an empty text embedding. The final noise prediction used for denoising is then a weighted average of these two predictions:

$$\tilde{\varepsilon}_\theta(z_t, t, C, \varnothing) = \omega \cdot \varepsilon_\theta(z_t, t, C) + (1 - \omega) \cdot \varepsilon_\theta(z_t, t, \varnothing). \quad (3)$$

The weighting factor $\omega$, known as the guidance scale, determines how much the final prediction is influenced by the text condition versus the unconditional generation.

**DDIM Inversion:** DDIM Inversion is a complementary technique that allows for the reconstruction of the initial noise latent from an existing image. This capability enables the editing of generated images by manipulating the recovered noise latent.

DDIM Inversion takes the final noisy image encoding $z_T$ and works backwards through the denoising steps to recover the initial noise encoding $z_0$ that was used to generate that image, which can be formulated as:

$$z_{t+1} = \sqrt{\frac{\alpha_{t+1}}{\alpha_t}} z_t + \left( \sqrt{\frac{1}{\alpha_{t+1}} - 1} - \sqrt{\frac{1}{\alpha_t} - 1} \right) \cdot \varepsilon_\theta(z_t, t, C). \quad (4)$$

Here, $z_t$ represents the noise encoding at each step $t$, $\alpha_t$ is a noise scaling factor, and $\varepsilon_\theta(z_t, t, C)$ is the noise prediction made by the diffusion model, conditioned on the current encoding $z_t$, timestep $t$, and any relevant conditioning information $C$ (e.g. a text prompt).

**Attention Control** The attention control mechanism, proposed by Prompt-to-Prompt [17], aims to replace the attention maps in the diffusion process using the following formula:

$$Edit(M_t, M_t^*, t) := \begin{cases} M_t^* & \text{if } t < \tau \\ M_t & \text{otherwise,} \end{cases} \quad (5)$$

where $M_t$ is the original attention map, $M_t^*$ is the edited attention map, and $\tau$ is a timestamp parameter that determines the step until which the attention map replacement is applied. This soft attention constraint allows the method to preserve the original composition in the diffusion steps, while enabling more targeted editing.

## 3.2 Efficient Viewpoints Selection

Existing methods for generating geometric and material properties based on diffusion models typically use either a large range of random viewpoints (e.g., DreamFusion [34], Fantasia3D [3]) or fixed viewpoints (e.g., TEXTure [35], Instruct-Nerf2Nerf [15]). While effective, these approaches suffer from either excessive redundancy (random viewpoints) or lack of adaptability and reliability (fixed

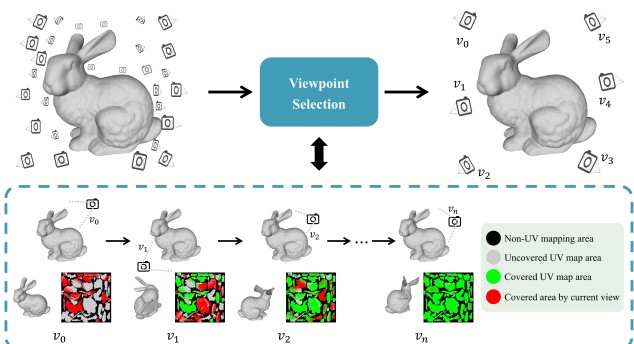

**Figure 3: Visualization of EVS's coverage. We render a UV color map from each selected viewpoint to show efficiency. After several selection, the viewpoints almost covered the entire surface of the bunny.**

viewpoints) for our translucent material transferring task. To address these issues and satisfy our requirement of covering the vast majority of the surface with only a few viewpoints. we propose a method called Efficient Viewpoints Selection (EVS), which consists of two steps:

**Viewpoint Sampling:** We first normalize the geometry of the mesh, then sample points on the mesh surface. For each sampled point, we define a corresponding viewpoint as the intersection between the surface normal at that point and a bounding sphere of radius 2 around the model. We set the camera orientation to point from each viewpoint towards its associated surface sample point and get sampled viewpoints $V_{sampled}$.

**Viewpoint Selection:** For each candidate viewpoint $v_i$ in $V_{sampled}$, we devide the surface of mesh into $S_{new}$ (new captured surface) and $S_{others}$ (uncaptured or already captured surface). Then we render an image which the the pixel of $S_{new}$ set to 1 and $S_{others}$ set to 0. We then evaluate the quality of each viewpoint using a scoring function that aims to maximize the coverage of surface at a fine angle. The scoring function is defined as:

$$Score(v_i) = \frac{\sum P_{value=1}}{A(S_{new})} \leftarrow, v_i \in V_{sampled}. \quad (6)$$

where $Score(v_i)$ is the score of the $i$-th viewpoint, $P_{(value=1)}$ is pixel with a value of 1 in rendered image (representing new captured surface), and $A(S_{new})$ is the total area of the new captured surface from the current viewpoint.

By calculating this scoring function for each candidate viewpoint and iteratively selecting the viewpoint with the highest score, the algorithm can efficiently choose a set of viewpoints $V_{selected}$ that cover nearly the entire surface of the 3D model, as demonstrated in Figure 3, the selected viewpoints after several iterations can cover nearly the entire surface of the bunny, demonstrating the effectiveness of our algorithm.

## 3.3 Translucent Differentiable Rendering

InverseTranslucent[5] introduced an end-to-end approach to simultaneously estimate the complex geometry and heterogeneous translucent properties of translucent objects from photographs

using inverse rendering techniques. This method uses a heterogeneous BSSRDF to represent translucency, and extends the framework of Path-Space Differentiable Rendering (PSDR) [46] to accommodate both surface reflection and subsurface scattering. The following integral formula is used to simulate the subsurface light transport problem:

$$L_o(\mathbf{x}_0, \omega_0) = \iint \rho_s(\mathbf{x}_o, \mathbf{x}_i, \omega_o, \omega_i) L_i(\mathbf{x}_i, \omega_i) |\cos \theta_i| \mathrm{d}A \mathrm{d}\omega_i$$
$$+ \int \rho_r(\mathbf{x_0}, \omega_0, \omega_i) L_i(\mathbf{x_0}, \omega_i) |\cos \theta_i| \mathrm{d}\omega_i. \tag{7}$$

Here, the outgoing radiance $L_o$ is the combined result of subsurface scattering and surface reflection. $\rho_r$ represents the BRDF model, calculating the surface reflection contribution at $\mathbf{x}_o$. $\rho_s$ represents the BSSRDF model, calculating the subsurface light transport contribution from incident light at $\mathbf{x}_i$ in direction $\omega_i$ to the outgoing direction at $\mathbf{x}_o$. The BRDF model describes the scattering behavior of light on rough surfaces, parameterized by the GGX distribution with surface roughness $\beta$ and refractive index $\eta$. For the BSSRDF model, the practical dipole model proposed in [19] is used to compactly represent homogeneous subsurface scattering materials. Spatially-varying parameters following [38] are introduced to model heterogeneity, with parameters including scattering albedo $\alpha$ and extinction coefficient $\sigma_t$. These physical parameters can be represented either as single values or as textures, collectively referred to as the parameter vector $\pi$.

To optimize the parameter vector $\pi$, the loss function $g(\mathbf{I}(\pi))$ is minimized. To effectively optimize the model in the presence of Monte Carlo noise introduced by the BSSRDF integral, a dual-buffer method is employed to evaluate L2 image loss:

$$g(\mathbf{I}(\pi)) = (\mathbf{I}_1(\pi) - \mathbf{I}_{\mathrm{ref}})(\mathbf{I}_2(\pi) - \mathbf{I}_{\mathrm{ref}}). \tag{8}$$

This provides an unbiased estimate of the difference between the rendered image $\mathbf{I}(\pi)$ generated based on parameters $\pi$ and the reference image $\mathbf{I}_{\mathrm{ref}}$. It effectively computes the loss value based on two Monte Carlo estimates, helping to reduce potential gradient estimation bias caused by correlations between a single rendering and its derivatives, even when using low sampling rates to ensure correct convergence of the optimizer.

In contrast to InverseTranslucent [5], our optimization process fixed the 3D model's geometric parameters and updated only the material-related parameters. We employed the translucent differentiable renderer to perform rendering and inverse rendering.

### 3.4 Translucent Style Image Transfer

**Pivot Turning Inversion:** A key challenge with standard DDIM inversion is the accumulated error when using CFG in text-guided diffusion models. The CFG guidance scale $\omega$ amplifies this error, leading to visual artifacts. The key idea of Pivot Turning Inversion (PTI) is to modify the unconditional embedding $\varphi_t$ associated with each timestamp $t$ to better match the given image and reduce error.

PTI inverts the initial image $x_{v_i}^{ini}$ and the current rendered image $x_{v_i}^{cur}$ in viewpoint $v_i$. Taking the inversion of $x_{v_i}^{cur}$ as an example: PTI optimizes the unconditional embedding $v_t^{cur}$ at each timestamp $t$ to minimize the distance between the ground-truth noise-free latent $z_0^{cur}$ and the denoised latent $\hat{z}_0(z_t^{cur}, v_t^{cur})$ estimated by the

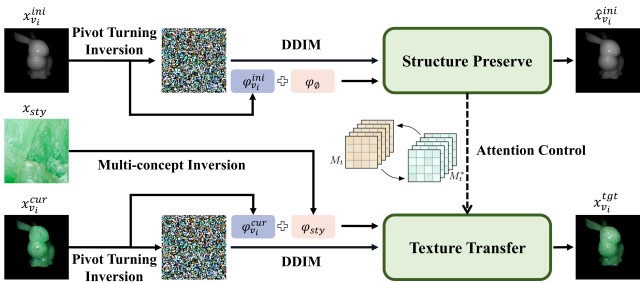

**Figure 4: Our structure-preserving translucent editor for image editing. The editor transfers the style image to the current rendered image while preserving the structure and lighting conditions of the initial image.**

pre-trained model:

$$\min_{v_t^{cur}} \|z_0^{cur} - \hat{z}_0(z_t^{cur}, v_t^{cur})\|,$$
$$\min_{v_t^{ini}} \|z_0^{ini} - \hat{z}_0(z_t^{ini}, v_t^{ini})\|. \tag{9}$$

Here, $z_0^{cur}$ is the noise-free latent of the editing image $x_{v_i}^{cur}$, and $\hat{z}_0(z_t^{cur}, v_t^{cur})$ is the denoised latent estimated by the pre-trained model. The same process is applied to $x_{v_i}^{ini}$. This approach learns an unconditional embedding that can perfectly reconstruct the inverted image with the initial noise latent.

**Multi-concept Inversion:** Multi-concept Inversion (MCI) focuses on learning a conditional embedding $\varphi_{sty}$ that extracts rich semantic information from the style image $x_{sty}$. However, the negative embedding used in prior methods like Textual Inversion[10] and DreamArtist [7] are not necessary in our case, as they may conflict with the unconditional embedding learned through PTI.

Therefore, we adopt a single positive multi-concept embedding approach used by VCT [4]. We fix the parameters of the pre-trained diffusion model and optimize the style embedding $\varphi_{sty}$ to minimize the following objective:

$$\mathcal{L}_l = E_{\epsilon,t} \left[ \left\| \epsilon - \epsilon_\theta(z_t^{sty}, t, \varphi_{sty}) \right\|_2^2 \right]. \tag{10}$$

Here, the style embedding $\varphi_{sty}$ represents the embedding of the style image $x_{sty}$, and $z_t^{sty}$ is the noisy latent of the style image at each timestamp $t$. The goal of this objective is to learn the style embedding that best predicts the noise residual $\varepsilon$, effectively capturing the essential visual concepts in the style image, which can aid our transfer process.

**Structure-preserving Translucent Editor:** The proposed structure-preserving translucent editor aims to transfer the style image to the current rendered image while preserving the structure and lighting conditions of the translucent initial image. The dual stream denoising architecture employed by VCT [4], which, although effective in first editing, resulted in unstable style transfer and loss of translucency and structural information during iterative editing.

To address these issues, we propose a structure-preserving translucent editor illuminated in Figure 4. The editor utilizes two branches:

Structure preserving Branch: We use PTI to invert the initial translucent image $x_{v_i}^{ini}$ of viewpoint $v_i$, obtaining its text embedding $\varphi_{v_i}^{ini}$ and initial noise latent $z_T^{ini}$, which are used to perfectly reconstruct the translucent initial image. During the reconstruction

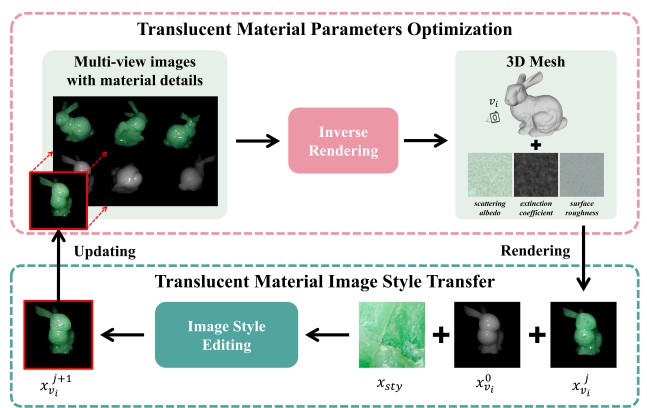

**Figure 5: Consistent Translucent Material Update.** Our iterative editing-optimization strategy ensures consistency in material editing for heterogeneous translucent materials across multiple viewpoints.

process, we extract the attention maps $M_t^*$ at each timestamp $t$. The denoise process of the structure preserving branch $B^*$ is as follows:

$$\tilde{\epsilon}\theta(z_t^{ini}, t, \varphi_{v_i}^{ini}, \varphi_\varnothing) = \omega \cdot \epsilon_\theta(z_t^{ini}, t, \varphi_\varnothing) + (1-\omega) \cdot \epsilon_\theta(z_t^{ini}, t, \varphi_{v_i}^{ini}), \quad (11)$$

where $\varphi_\varnothing$ is the empty text embedding, and $\omega$ is the guidance scale for CFG.

Style Transfer Branch: We also use PTI to invert the current rendered image $x_{v_i}^{cur}$ of viewpoint $v_i$, obtaining its text embedding $\varphi_{v_i}^{cur}$ and initial noise latent $z_T^{cur}$. Meanwhile, we use MCI to invert the style image $x_{sty}$ and obtain the style embedding $\varphi_{sty}$. We then use the two learned embeddings as conditions to denoise the noise latent in the style transfer branch $B$:

$$\tilde{\epsilon}\theta(z_t^{cur}, t, \varphi_{v_i}^{cur}, \varphi_{sty}) = \omega \cdot \epsilon_\theta(z_t^{cur}, t, \varphi_{sty}) + (1-\omega) \cdot \epsilon_\theta(z_t^{cur}, t, \varphi_{v_i}^{cur}), \quad (12)$$

note that the $\omega$ in the above two formulas is the same.

During the simultaneous denoising process described above, we use the attention control mechanism in Eq.5, extracting $M_t^*$ from $B^*$ to replace $M_t$ of $B$. This ensures that the structure and light conditions of the initial image are perfectly preserved during the style transfer.

The editing process is as follows:

$$B^* : z_T^{ini} \rightarrow z_{T-1}^{ini} \rightarrow \cdots \rightarrow \hat{z}^{ini}$$
$$B : z_T^{cur} \rightarrow z_{T-1}^{cur} \rightarrow \cdots \rightarrow z_{tgt}^{cur} \quad (13)$$

We continuously render-edit-optimize, and during the editing process, the structure preserving branch always uses the initial image $x_{v_i}^{ini}$, while the style transfer branch's input image iteratively changes from current rendered $x_{v_i}^{cur}$, and the edited result $x_{v_i}^{tgt}$, gradually achieving global consistency with the style image across multiple viewpoints.

### 3.5 Consistent Translucent Material Update

Editing the rendered translucent images only once may generate good results, but could lead to inconsistencies across multiple viewpoints, where some edited images do not perfectly transfer the style and remain close to the originals. Instruct-Nerf2Nerf [15] adopts

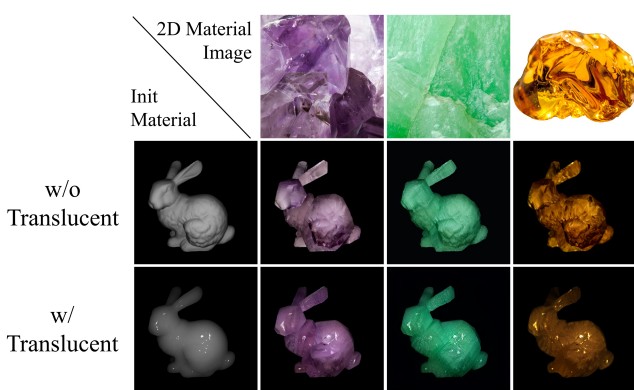

**Figure 6: Material Initialize.** We present edited results without translucent material initialization under different style images and compare them with our results. In contrast, our edited results with translucent material initialization better preserve translucent details.

an iterative editing approach, alternately updating the dataset to achieve consistent NeRF scene editing. Inspired by this alternating update strategy, our method uses iterative editing to update the translucent dataset. Our structure-preserving translucent editor ensures stability across multiple edits, preventing "drift" and loss of structure and translucency information during the iterative material updates. Figure 5 shows our iterative editing-optimization strategy for heterogeneous translucent materials across multiple viewpoints.

**Translucent Material Initialization:** Material initialization is crucial as it determines the optical information. As most semantic information is concentrated in non-black areas, and higher color values are more easily perceived by the attention mechanisms of diffusion models, our translucent initialization meets the requirement of preserving only structure and lighting for material transfer. Opaque materials are unable to capture translucency details due to the lighting model's inability to describe subsurface scattering. As demonstrated in Figure 6, without translucent initialization, edited results clearly lose translucency details under the same lighting and viewing conditions compared to renderings with translucent initialization, which achieve high-quality texture transfer while preserving initial lighting. The detail of our translucent initialization is in Section 4.1.

The translucent dataset, denoted as $D_T$, is designed to facilitate the transfer of materials from 2D to 3D. Initial rendered images $x_{v_i}^0$ are rendered for each viewpoint, and this collection forms the initial translucent dataset $D_T$.

**Iterative Translucent Dataset Update:** After initializing the translucent dataset $D_T$, an iterative update process is performed, alternating between the translucent style image style transfer $E$ and the translucent material parameters optimization $O$.

In process $E$, the current image $x_{v_i}^j$ in $j$-th iteration is rendered and edited to obtain $x_{tgt}^j$, which updated the $D_T$. During the process, $D_T$ transitions from the old state to the new state. Next editing process will edit current image rendered in next view ($v_i \rightarrow v_{i+1}$).

In process $O$, an image from $D_T$ is randomly selected for supervision, and the translucent parameters are optimized through a differentiable renderer using inverse rendering.

The combination of the $E$ and $O$ allows for the gradual refinement of $D_T$. By alternating between these processes, the material properties are iteratively updated to match the desired style while maintaining consistency across different viewpoints. This iterative editing-optimization strategy ensures consistent material editing for heterogeneous translucent materials across multiple viewpoints.

## 4 RESULTS

Figure 1 shows the results of our method of editing various standard models with different style images. The selected results can be found in the appendix. Our method accomplishes numerous challenging edits, including those involving semitransparent materials like jade and marble found in nature. We elaborate on the implementation details of our method in Section 4.1. In Section 4.2, we qualitatively compare our approach with renowned works such as $StyTR^2$ [5], Artflow [23], and InstructPix2Pix [2] in image-to-image (I2I) tasks. To validate our method, we conducted ablation experiments against a set of ablative baselines in Section 4.3.

### 4.1 Implementation details

Our work was conducted with the following specifications: The translucent renderer was performed at a resolution of [512, 512] pixels, and the field of view (FOV) was set to 45 degrees. In the rendered scene, a pinhole camera was placed on a sphere with a radius of 2, and a point light was utilized, positioned around the camera at a distance of 0.2. The power of the point light was set to 20000. To enhance rendering precision, the entire scene was scaled by a factor of 10.

For our translucent material initialization, the extinction coefficient $\sigma_t$, which describes light attenuation through the medium, was initialized to [1.5, 1.5, 1.5], a value close to that commonly observed in translucent materials. Higher values of $\sigma_t$ indicate stronger absorption and scattering, resulting in faster decay. The scattering albedo $\alpha$, which represents the probability of light scattering at a location in the medium, was initialized to [0.9, 0.9, 0.9]. All these parameters were presented in the form of textures with a resolution of [512, 512] pixels.

A total of 500 iterations were conducted during the optimization process, which took approximately 18 minutes on two NVIDIA 3090 GPUs. For the optimization process, the same loss function as InverseTranslucent [5] was used, along with an AdamW optimizer [28]. After performing 50 optimization iterations, an editing process was conducted. For the editing process, the style image embedding was initially trained for 500 iterations. During the inference stage, a cross-attention ratio of 0.2 and a self-attention ratio of 0.9 were used.

### 4.2 Baseline comparisons

**Comparison with image editing models.** Figure 7 shows the result of a comparison between our methods and the baseline image editing methods. As for baselines, we selected some state-of-the-art methods, including InstructPix2Pix, $StyTR^2$, and Artflow. The results indicate that our approach can edit images while preserving

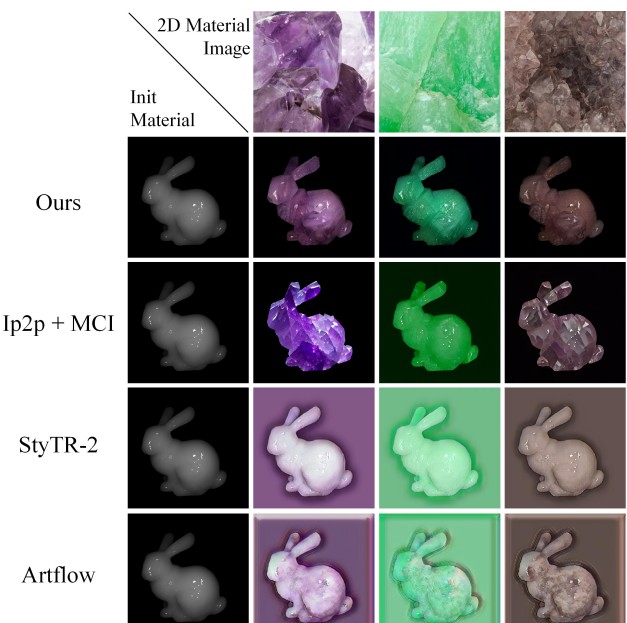

**Figure 7: Comparison with other image editing methods. Our method accurately transfers the material while preserving the translucent information of the material. More importantly, the edited results also elegantly maintain the geometric shapes and lighting features of the models in the images.**

the translucent information of the material, as well as the geometric structure of the model and lighting features. Specifically, in the results from $StyTR^2$, the image editing outcome exhibits colors similar to the style image but loses a significant amount of texture features and translucent information. The results in Artflow retain more texture features from the style image compared to $StyTR^2$ but still fall short of the desired effect. In the results of InstructPix2Pix with Multi-concept inversion, the method effectively preserves the texture details from the style image by employing multi-concept inversion to generate concept embeddings representing complex visual concepts. Moreover, the results demonstrate the ability to retain translucent information and lighting features. However, regrettably, this method loses the geometric structure of the model in the image.

**Comparison with 3D editing models.** We compare our method with other 3D model editing approaches. As for baselines, we selected prominent methods in 3D scene editing, including Instruct-NeRF2NeRF [15]. Figure 8 shows the results of our method alongside those of the baseline methods on 3D models. NeRF2NeRF performs editing on NeRF scenes guided by text instructions, but its output exhibits limitations in expressing surfaces with complex materials. In an effort to improve the outcomes, we substituted the image editing method in Instruct-Pix2Pix [2] from NeRF2NeRF with other image editing techniques. However, since NeRF2NeRF is primarily trained on 3D NeRF scenes, it is evident that the baseline methods based on NeRF2NeRF and its variants still encounter difficulties in editing the translucent materials. Nevertheless, our method successfully transfer the style image to the 3D model's translucent material with consistence.

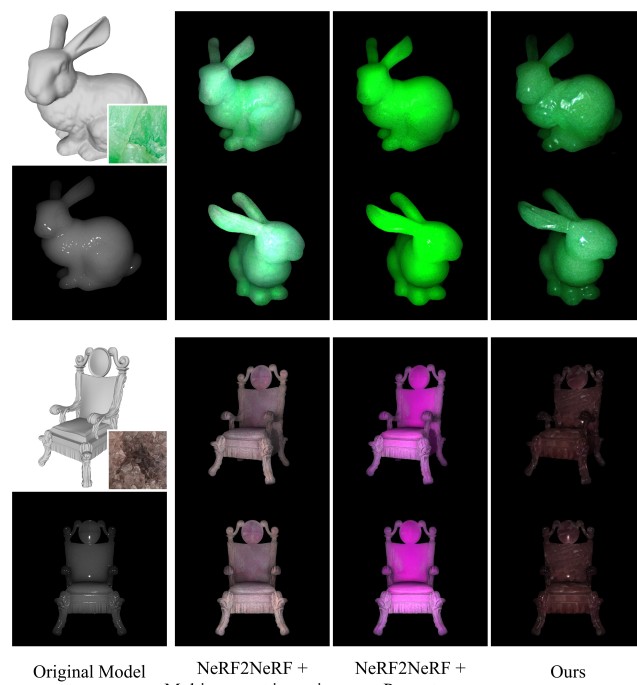

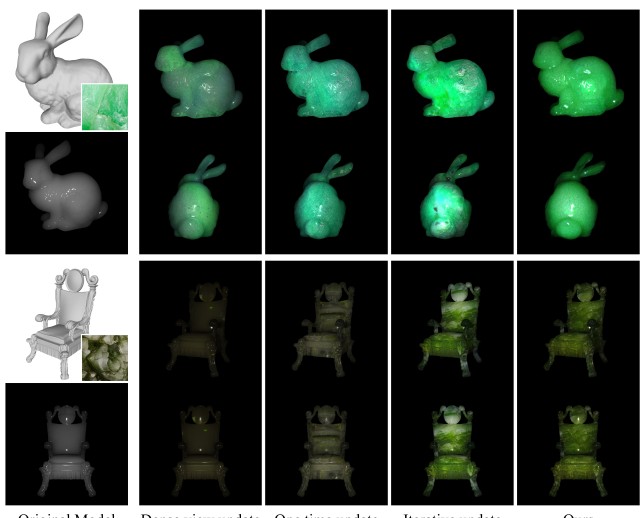

| Original Model | NeRF2NeRF + Multi-concept inversion | NeRF2NeRF + Prompt | Ours |

Figure 8: Comparison with other 3D scene editing methods. Comparison with variants of various 3D scene editing methods. The bunny and chair models are edited using images with jade and apophyllite materials. Our results effectively preserve the texture details and translucent information of the style images.

### 4.3 Ablation

In order to verify the necessity of each component in our method, we conducted ablation studies. The qualitative differences are shown in Figure 9:

**Dense view update.** In this baseline, we do not employ our efficient viewpoint selection method to select high-quality viewpoints. Instead, we obtain dense and uniform viewpoints around the model. Images from all viewpoints are edited only once, and all participate in iterative training. From the results, dense viewpoints lead to the averaging of texture features in the final iteratively edited outcome, causing a loss of material features.

**One time update.** The next method adopts a strategy of updating the dataset only once. In this baseline, we first use the efficient viewpoint selection method to choose good viewpoints, but during iterative dataset updates, we edit the rendering images of each viewpoint only once. Results indicate that a single edit is not sufficient to extract the features from the style image.

**Iterative update.** Building upon the One-time update approach, we adopt the strategy from Instruct-NeRF2NeRF [15] to iteratively edit the translucent dataset. However, during the editing process, we refrain from using the initial image. Although this method can produce decent editing results, the outcome lacks a significant amount of translucent information in the absence of the initial image. This experiment highlights the importance of the structure preserving branch of our structure-preserving translucent editor.

| Original Model | Dense view update | One time update | Iterative update | Ours |

Figure 9: Ablation study results. We compare our method with a collection of variants described in Section 4.3. Dense view update shows the results of editing with uniformly dense viewpoints; One-time update presents the outcome of a single-time editing strategy; Iterative update displays the results of iterative editing without utilizing initialized translucent input.

## 5 CONCLUSION

This paper introduces an innovative method for estimating parameters of heterogeneous translucent materials from a single image, enabling direct parameter estimation under natural lighting conditions and material transfer onto a 3D model. Our contributions encompass a novel single-image-based transfer method for heterogeneous translucent materials, a translucent material transfer editor preserving lighting and structure, and an iterative editing approach ensuring consistency across multiple viewpoints. Despite the practical and accessible nature of our approach, overcoming limitations of existing methods, there are some constraints. Recovering the geometry and appearance of translucent objects from sparse views under strong illumination remains a challenge, requiring further improvement in our method's learning capability for materials occluded by highlights. Additionally, our algorithm currently lacks support for scene-level 3D editing under ambient lighting. Despite these limitations, we believe our method marks a significant step in 3D object material editing, offering new possibilities for 3D modelers and artists to efficiently transfer translucent materials onto 3D models.

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
