# OpenReview forum: "Learning to Transfer Heterogeneous Translucent Materials from a 2D Image to 3D Models"
_acmmm.org/ACMMM/2024/Conference — MM2024 Poster_

### Official Review · Reviewer_T9fk · 2024-05-09

**Rating:** 4
**Confidence:** 3

**Summary:**

This paper is about rendering translucent material to 3D models. They present a new approach for estimating and transferring the parameters of heterogeneous translucent materials from a single 2D image to 3D models. The contribution of the methods contains a method for transferring heterogeneous translucent materials based on a single image. A translucent material transfer editor preserving
 both lighting and structure. An iterative editing-optimization strategy. The experiment results shows it has good performance.

**Strengths:**

1. Compared to the previous method, the rendering performance of the proposed method is absolutely better and the results are convincing.
2. For the experiment results, the model proposed in this paper outperforms the baseline model and the paper analyzes enough numbers of evaluation metrics.
3.  It is a well-formatted thesis. It has proper headings, figures, tables, and illustrations, making it easy to read and follow.
4. The contribution of the methods is solid.

**Limitations:**

1. The experiment results are not enough. There are only a few number of rendering results. It makes readers worried about the generalization performance.
2. There are no quantitative results. Maybe the author can seek some quality assessment methods to evaluate the quality of the generated results and compare them to other methods.

**Suitability:**

2

---

### Official Review · Reviewer_HcHp · 2024-05-16

**Rating:** 3
**Confidence:** 2

**Summary:**

This paper proposes a method for transferring heterogeneous translucent materials from a single image. Their method enables the direct estimation of relevant parameters under natural lighting conditions and seamlessly transfers the material onto a 3D model. To achieve this objective, they introduce an innovative iterative editing-optimization strategy, which iteratively edits and performs inverse rendering. To achieve this objective, they introduce an innovative iterative editing optimization strategy, which iteratively edits and performs inverse rendering. To ensure consistency and coherence in our multi-view material editing results, they update the edited translucent results onto the 3D model.

**Strengths:**

1. The description of the paper is clear and easy to understand.
2. This paper proposes a novel method to direct the estimation of relevant parameters under natural lighting conditions and seamlessly transfer the material onto a 3D model.
3. This paper proposes an iterative editing-optimization strategy to ensure consistency in material editing for heterogeneous translucent materials across multiple viewpoints.
4. The Efficient Viewpoints Selection (EVS) is interesting for the viewpoint chosen.

**Limitations:**

1. The paper has few experiments; it is recommended to include more comparative experiments to verify the validity of the method.
2. The paper does not conduct a quantitative analysis of the experiment。

Further comments:
1. What results will it produce for shapes with more complex topology? For example, shapes containing more genus.
2. Can multi-modal information be added to enhance effect and controllability?

**Suitability:**

2

---

### Official Review · Reviewer_Amk1 · 2024-05-22

**Rating:** 4
**Confidence:** 1

**Summary:**

To automatically estimate the parameters for heterogeneous translucent materials, this paper proposes a single-image based method that includes viewpoint selection, translucent material initialization, and an iterative editing-optimization strategy.

**Strengths:**

- This paper proposes a workflow where each stage is carefully designed. The effectiveness is proved by a lot of solid visual results presented in the paper.
- The structure-preserving translucent editor integrates components from the latest progress in image generation while faithfully preserving the structure and lighting, overcoming the drawbacks of the traditional style-transfer methods.

**Limitations:**

- The organization of some sections is a little messy. E.g., Sec 3.3 and the PTI part of Sec 3.4 mainly explain the prior methods, so they may be moved to the preliminaries. And Fig 6 should be moved to the ablation study.
- It’s hard to figure out process E and process O mentioned in Sec 3.5. It would be better to visualize them for better understanding.
- The conclusions would be more convincing if more quantitative results are shown in the paper: e.g., some metrics to compare the texture similarity between the results and the reference image; and the accuracy of the estimated material parameters.

**Suitability:**

3

---

### Official Review · Reviewer_2GbU · 2024-05-25

**Rating:** 4
**Confidence:** 2

**Summary:**

This paper proposes a new approach to estimate the parameters of heterogeneous translucent materials from a single 2D image and transfer them to a 3D model.

**Strengths:**

The paper is well structured and the experimental results are superior to the existing SOTA.

**Limitations:**

1) In the paper, the method is described as minimizing redundancy, but isn't some degree of redundancy required in 3D modeling processing? Is it used to validate performance by cross checking redundant areas or not?

2) Is it unavoidable that the result of StyTR-2, Artflow in Figure 7 has a different color background? It looks like it has a worse effect due to the background. Compared to the other two, StyTR-2 seems to be targeting relatively similar translucent materials. The results in the StyTR-2 paper look relatively good compared to the results shown in Figure 7. Is there a difference between the experimental environment in StyTR-2 and this paper? Is it a difference in samples? Is it possible to do a comparison with the samples used in the StyTR-2 paper?

3) Modeling performance for translucent materials is not only about the results from a single viewpoint, but also about the naturalness of the change as the viewpoint is shifted, what are the results? I know it's hard to show this in the paper, but maybe you could show it in a video, or maybe a metric to compare the degree of naturalness.

4) I understand the difficulty, but is there any way to show an objective comparison? I think it is ambiguous to judge what is better just by looking at the images in Figure 9 and the results in the supplementary document.

5) One of the main contributions of the paper is efficient viewpoint selection, and it's a shame that the results are missing. There is a lack of experiments that show the results of efficient viewpoint selection. For example, experiments on what performance reduction is caused by excessive viewpoint selection, etc.

6) Other minor comments include the following.
- The text in Figure 2 is too small and hard to read.
- Page 4, right column, lines 438, 439, font style error.
- Page 4, right column, line 440, typo “the the”.

**Suitability:**

3

---

### Meta-Review · Area_Chair_GG3o · 2024-06-27

**Recommendation:** Accept (Poster)
**Confidence:** 4

**Metareview:**

This paper received four reviews. Three reviewers are positive about this paper. They consider that the workflow is well designed and the results are strong. The negative reviewer is concerned with the lack of quantitative results and the experiments are not sufficient. The rebuttal provides some quantitative results. Given the overall positive feedback received by this paper, we are happy to recommend the acceptance of this paper.